# U-Net with Asymmetric Convolution Blocks for Road Traffic Noise Attenuation in Seismic Data

Zhaolin Zhu [1] , Xin Chen [2,*], Danping Cao [2], Mingxin Cheng [1,3] and Shuaimin Ding [1,3]

1 Hainan Institute of Zhejiang University, Sanya 572024, China
2 School of Geosciences, China University of Petroleum (East China), Qingdao 266580, China
3 Advanced Technology Institute of Zhejiang University, Hangzhou 310027, China
* Correspondence: xinchen@s.upc.edu.cn

**Abstract:** Road traffic noise is a special kind of high amplitude noise in seismic or acoustic data acquisition around a road network. It is a mixture of several surface waves with different dispersion and harmonic waves. Road traffic noise is mainly generated by passing vehicles on a road. The geophones near the road will record the noise while receiving the seismic signal. The amplitude of the traffic noise is much larger than the signal, which masks the effective information and degrades the quality of acquired data. At the same time, the traffic noise is coupled with the effective signal, which makes it difficult to separate them. Therefore, attenuating traffic noise is the key to improving the quality of the final processing results. In recent years, denoising methods based on convolution neural networks (CNN) have shown good performance in noise attenuation. These denoising methods can learn the potential characteristics of acquired data, thus establishing the mapping relationship between the original data and the effective signal or noise. Here, we introduce a method combining UNet networks with asymmetric convolution blocks (ACBs) for traffic noise attenuation, and the network is called the ACB-UNet. The ACB-UNet is a supervised deep learning method, which can obtain the distribution characteristics of noise and effective signal through learning the training data and then effectively separate the two to achieve noise removal. To validate the performance of the proposed method, we apply it to synthetic and real data. The data tests show that the ACB-UNet can obtain good results for high amplitude noise attenuation and is practical and efficient.

**Keywords:** road traffic noise; high amplitude; UNet; asymmetric convolution blocks



## 1. Introduction

Seismic or acoustic wave signals are inevitably contaminated by noise from various sources in different environments, such as land and marine, affecting the quality of imaging, inversion, and interpretation [1–4]. Hence, denoising is one of the most critical processing steps in oil and gas exploration or engineering survey, and it can significantly enhance the signal-to-noise ratio (SNR) and resolution of real data. Ground roll traffic noise in land surveys and swell noise in marine acquisition are very important ambient noises which have a high amplitude and mixed models. These noises are mixed with valid signals, making seismic records more chaotic, and the energy of noise is high, often covering an effective signal, reducing the quality of seismic data. Attenuating the noise in seismic records and improving the signal-to-noise ratio can make more effective use of seismic data and facilitate subsequent inversion, imaging, and other operations.

In this article, we analyze the characteristics of road traffic noise and study how to attenuate traffic noise in seismic records to obtain effective signals. When sensors are deployed near a highway, the acquired data will be seriously polluted by traffic noise, which is the mixture of harmonic noise and ground roll from passing vehicles. Traffic noise belongs to a high-amplitude noise, and its characteristic is that its absolute amplitudes are more significant than most signals. Many kinds of literature have shown different ideas for suppressing high-amplitude noise effectively. Guo and Lin presented two techniques to solve

the coherent and non-coherent high-amplitude noise attenuation [5]. Schonewille et al. proposed an iterative F-X prediction filtering approach to attenuate swell noise [6]. Bekara and Baan proposed an automatic threshold-determination technique for detecting high-amplitude noise using statistical modeling [7]. As a kind of high-amplitude noise, road traffic noises are rarely mentioned in the literature, but they are very common in urban seismic or acoustic acquisition. The frequency band and amplitude level of traffic noise are comparable to data and are coupled with effective signals, so it is difficult to remove them. The F-X prediction method or the simple trace editing can suppress them in land data, but the methods are difficult for satisfying the demands of high precision. Domain transformation is required for noise removal by f-k filtering, Radon transformation, and other methods. Due to algorithmic mechanisms, these methods can generate artifacts that affect the use of effective signals. At the same time, in the f-k domain or Radon domain, effective signals and noise are often mixed together, making it difficult to completely separate. Traditional methods can cause the loss of effective signals when attenuating traffic noise. Wu et al. introduce $l_p - norm$ in the RPCA to effectively remove strong traffic noise, but the choice of parameter $p$ in $l_p - norm$ requires repeated experiments to achieve the best performance [8]. The traditional method requires human setting of parameters when attenuating noise. The increasing volume of seismic data and human manipulation not only affects efficiency, but also generates subjective errors. To overcome the shortcomings of traditional denoising methods, we introduce deep learning methods to more intelligently and efficiently attenuate traffic noise in seismic data.

With the development of deep learning research in geophysics, deep learning methods are used to first break picking [9,10], seismic data reconstruction [11,12], inversion [13–15], noise attenuation [16–22], etc. The clever and automatic noise attenuation technique based on the deep neural network was studied as an essential direction; multi-layer network structures and nonlinear activation functions can solve complex nonlinear problems. Unlike traditional methods, the deep learning method can be trained to extract the potential features of wave data through many hyperparameters without manual adjustment [23], which has more effective noise attenuation performances for real data. Tibi R et al. used the trained deep CNN model to decompose an input waveform into a signal of interest and noise; test results suggest that most of the recovered signal waveforms have high similarity to the target waveforms and suffer little distortion [24]. Saad and Chen proposed a deep-denoising autoencoder (DDAE) to attenuate random noise, and the DDAE encodes the input data to multiple levels of abstraction. Then, it decodes those levels to reconstruct the signal without noise [25]. Wang and Chen trained a deep CNN with residual learning for denoising to avoid artifacts generated in the results by transform-based denoising methods [26]. However, the deep learning method has yet to apply to traffic noise attenuation. In the seismic or acoustic record, traffic noise shows as vertical stripes with higher amplitudes, which arise earlier than the first arrival and grow stronger with time. These features can be identified and utilized by using the deep learning method. We introduce the UNet to attenuate the traffic noise. The network improves the working of convolutional neural networks significantly by combining the down-sampling path and up-sampling path with skip connections. In recent years, the UNet has been widely used in seismic or acoustic data processing and interpretation [27–34].

Although the UNet method is superior to the traditional method, its feature extraction ability still has room for improvement. Ding et al. (2019) proposed ACBs to replace the original square convolution blocks in the convolution network to enhance the performance of CNN [35]. To improve the ability of the UNet to separate traffic noise and effective signals, we introduce ACBs and propose an improved model termed UNet with ACBs (ACB-UNet). The denoising ability of the network is closely related to the quality of training data. To train the network more effectively and efficiently, a large amount of synthetic data and real data are used to construct the training data.

This paper introduces the ACB-UNet to suppress road traffic high amplitude noise. First, we introduce and analyze the acquired data set and traffic noise and generate the

training data of the network. Then, we present the architecture of the ACB-UNet and related theories. Finally, the ACB-UNet is applied to synthetic and real data, and the test shows that the road traffic noise can be attenuated very well.

## 2. Data Preparation

The data set studied in this paper is from land acquisition along the road. This record contains abundant high-amplitude traffic noise, which seriously degrades the quality of subsurface reflections. Therefore, we need to improve the useful signals by denoising, while maintaining the essential characteristics of the data profile to serve the subsequent data processing and interpretation.

### 2.1. Data Set

The real data are from land acquisition data of East China, which is corrupted by a lot of road traffic noises because a road network covers the survey. The acquisition geometry is shown in schematic form (Figure 1). The data set includes multiple ensembles, and 49 of them are extracted for the tests in this paper. Each ensemble has 360 receivers with an interval of 25 m, and its time samples are 3501 with the sampling. The simple pre-processing step, trace editing, is used for data before our tests, and the layout of noises is clear.

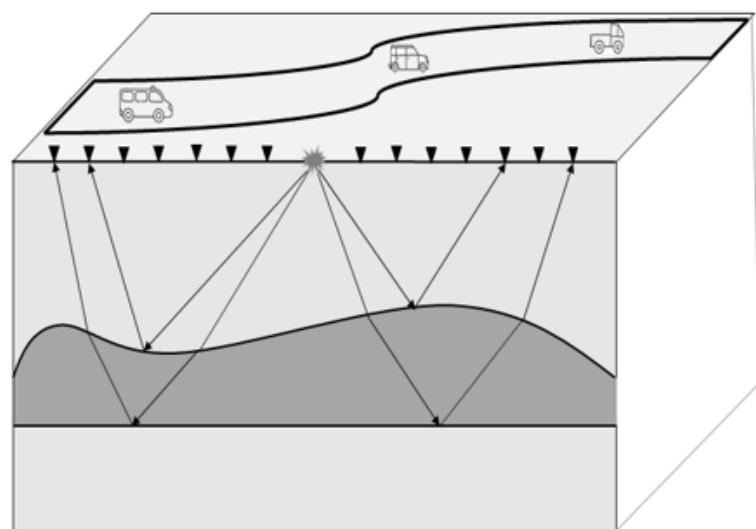

**Figure 1.** The land acquisition scenario of road noises.

### 2.2. Road Traffic Noises

Road traffic noise is characterized by high amplitude and is a mixture of several surface waves with different dispersion and harmonic waves. Figure 2a shows one ensemble of oil and gas exploration records (compressional data), including the road traffic noises, and the most apparent part of the road traffic noises in the black box of Figure 2a is magnified into Figure 2b. As can be seen from the figures, the traffic noise appears as vertical strips, appearing earlier than the first arrival and growing stronger with time. Meanwhile, the spectral analysis of data is also carried out. Figure 3a,b show the corresponding f-x and f-k spectra. In different transform domains, traffic noise is coupled with effective signals and noticeably interferes with reflection. It is noteworthy that traffic noise has other characteristics in different domains. In the f-x spectra, the noise is distributed in vertical bands, which is not significantly different from the effective signal. In the f-k spectra, the noise appears as horizontal bands, and the difference between it and the effective signal can be clearly observed. The characteristics of traffic noise provided the basis for the ACB-UNet to separate it from effective signals.

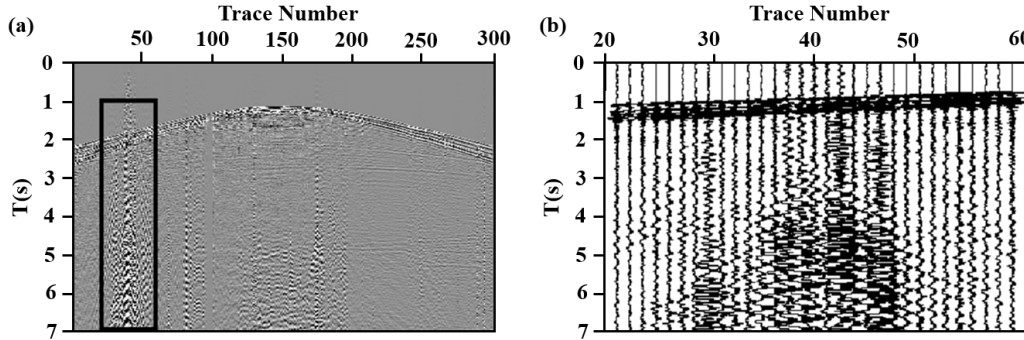

**Figure 2.** (**a**) The real data including road noises; (**b**) the main characteristics of the road noises zoomed from the black box.

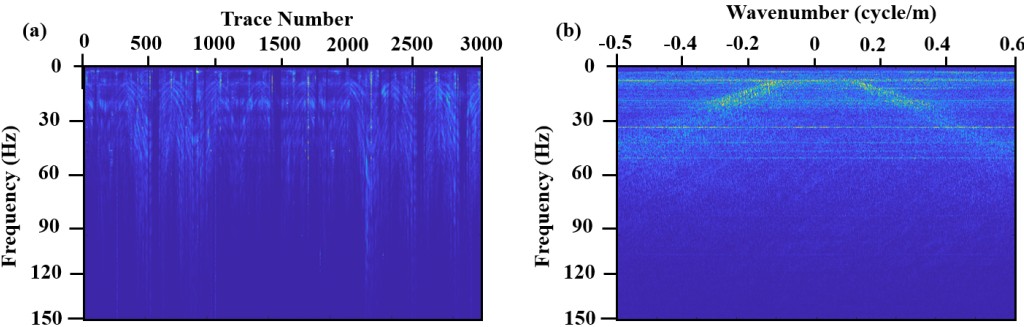

**Figure 3.** The real data in different transform domains: (**a**) f-x spectra; (**b**) f-k spectra.

### 2.3. Training Data and Parameter

The ACB-UNet effectively attenuates the traffic noise in acquired data relying on high-quality training data. We generated noise-free synthetic records through forward modeling based on the acoustic equation. Then, we added the road noise randomly to the synthetic records. The noise and the acquired data before and after adding noise construct the training pair of synthetic data. To increase the diversity and realism of the training samples, we processed the real data to obtain the data before and after denoising to enrich the training samples. Figure 4 shows the training sample pairs for the synthetic and real data. Before inputting the training samples into the ACB-UNet, the data need to be processed.

Since the amplitudes of traffic noise are larger than those of the reflections in data, to prevent the poor performance of the ACB-UNet, we need to normalize the data to the range of $[-1, 1]$, which is better adapted to the network. We can split the data into small patches to reduce computational time and memory consumption to train the network. It is feasible because there is no coherence between data signal and traffic noise. At the same time, data can be divided into small pieces to supply more training samples. In this paper, we choose the size of the patch as $128 \times 128$. It should be noted that there is some overlap among patches to solve boundary artifacts.

Appropriate parameters can help the network separate noise and effective signal, thus attenuating road noise. The initial learning rate of the network is 0.0002, and the Adam optimizer is used to optimize the network. The input and output sizes of the network are set to $128 \times 128$, and we set the batch size to 64.

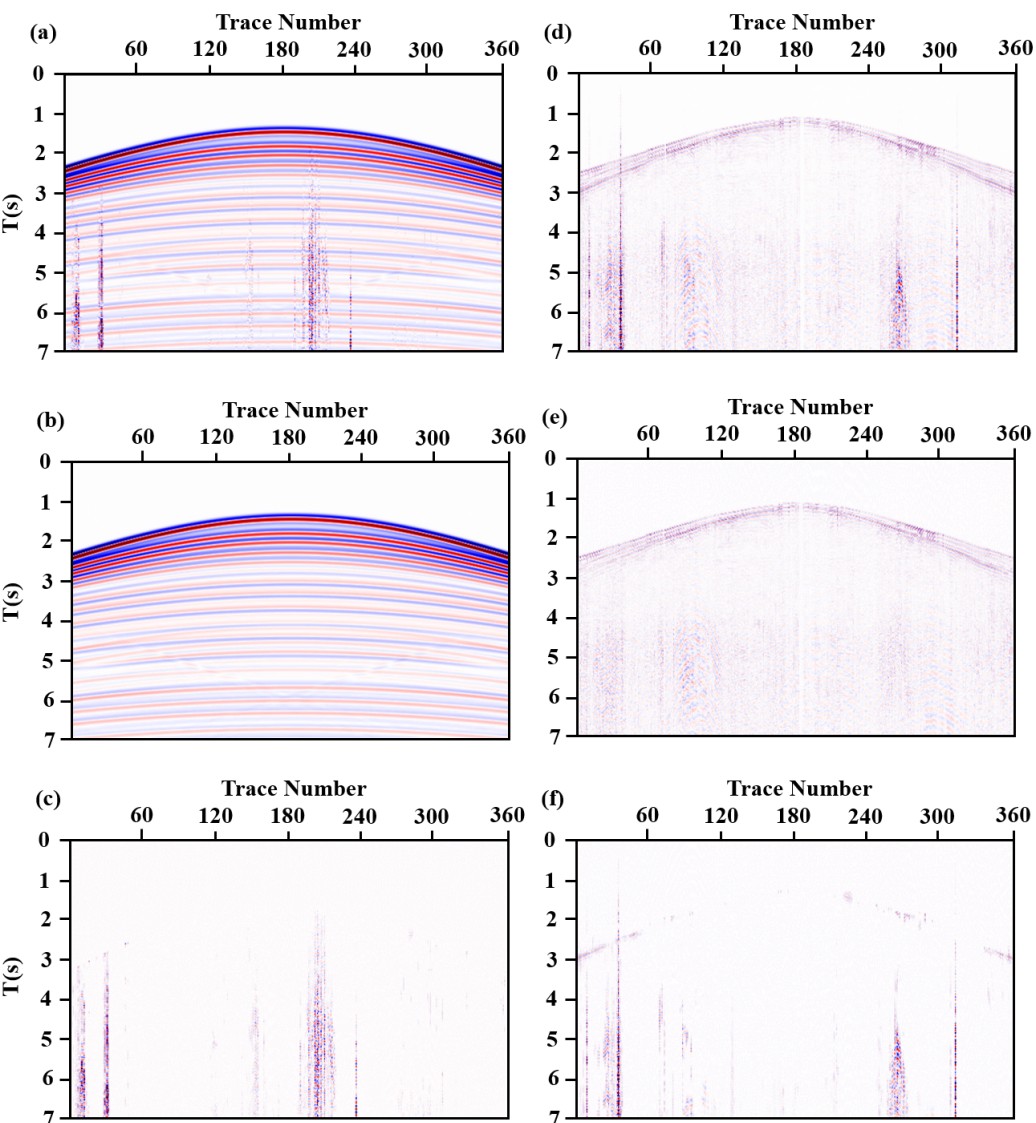

**Figure 4.** Training samples of synthetic data: (**a**) synthetic data with noise; (**b**) denoised synthetic data; and (**c**) traffic noise; training samples of real data: (**d**) real data with noise; (**e**) denoised real data; and (**f**) traffic noise.

## 3. Methods

Generally, the mixture model to describe the acquired data polluted by road traffic noises could be expressed as (1),

$$y = x + n \tag{1}$$

where $y$ represents acquired data with noise, $x$ represents the effective signals, and $n$ represents the traffic noises.

In Formula (1), the signal $x$ is not only expected to be recovered but also the noise $n$ should be separated from the original data. After the training, the ACB-UNet can construct the mapping between the original acquired data and the signals or noises. The mapping relationship can be written as (2),

$$(x^{pred}, n^{pred}) = N(y; \theta) \tag{2}$$

where $x^{pred}$ represents the estimated signal, $n^{pred}$ represents the estimated traffic noise, $N$ denotes the mapping function, and $\theta$ denotes network parameters. $N$ includes two elements: the first is the relationship from $y$ to $x^{pred}$; the second is from $y$ to $n^{pred}$.

### 3.1. UNet Structures

We adopt the UNet for the denoising network, originally proposed and successfully used for medical image segmentation [36] and then promoted and applied. The UNet is named for its U-shaped network architecture, as shown in Figure 5. It consists mainly of two paths, one is the encoder part, which is used to extract the imagery features, and the other is an expansion path, which operates the opposite way to restore the imagery features. The encoder includes two $3 \times 3$ convolutional layers, nonlinearity layers (ReLu), and $2 \times 2$ max pooling layers. On the decoder part, the structure is the opposite of the encoder. The decoder consists of two convolutional layers, each of which is followed by a ReLu and a transposed convolutional layer. The skip connections are added between the down-sampling path and the up-sampling path, which can combine local information with global information to avoid the vanishing gradient problem and improve network performance.

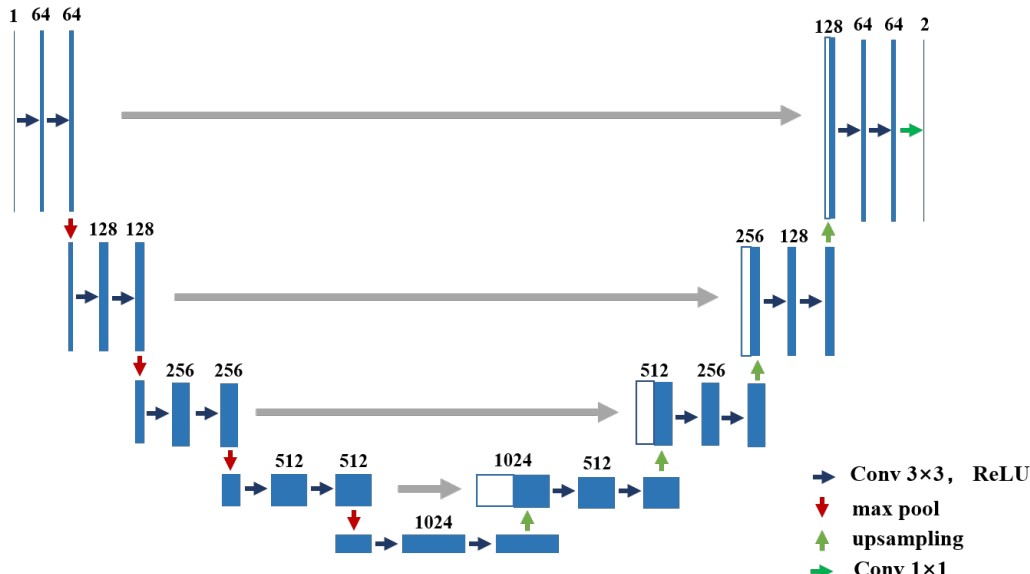

**Figure 5.** The architecture of the UNet.

In order to estimate the network parameters, an optimization problem will be solved as the following formula,

$$\hat{\boldsymbol{\theta}} = \arg \min_{\boldsymbol{\theta}} \frac{1}{M} \sum_{m=1}^{M} \mathcal{L}[\boldsymbol{N}(\boldsymbol{y_m}; \boldsymbol{\theta}, \boldsymbol{n}), \boldsymbol{x_m}, \boldsymbol{n_m}] + \lambda \|\boldsymbol{\theta}\|^2, \tag{3}$$

where $\{(\boldsymbol{y_m}; \boldsymbol{x_m}, \boldsymbol{n_m})\}$ is a training data set, $\boldsymbol{x_m}, \boldsymbol{n_m}$ are, respectively, a clean signal and mixed-model noise, and $\boldsymbol{y_m}$ is the polluted signal. The first term $\mathcal{L}[\cdot]$ in the above Equation (3), and the second term is the regularization term with a hyperparameter $\lambda > 0$.

### 3.2. Asymmetric Convolution Blocks

When using the UNet for road traffic noise attenuation, convolution is a pivotal step for feature extraction. The UNet commonly uses a $3 \times 3$ square convolution kernel for operation, and the operation process is shown in Figure 6. The process can also be expressed as follows (4):

$$\boldsymbol{y} = \boldsymbol{\omega} \times \boldsymbol{x} + \boldsymbol{b}, \tag{4}$$

where $\boldsymbol{x}$ is the input matrix, $\boldsymbol{y}$ is the convolutional result matrix, $\boldsymbol{\omega}$ is the weight of the convolution kernel, and the $\boldsymbol{b}$ is the bias. The training process of the network can be regarded as the optimization process of the convolutional kernel [37]. After training to find the optimal $(\boldsymbol{\omega}, \boldsymbol{b})$, the network can separate the effective signal and noise from the

original data. To further improve the feature extraction ability of the network, we use ACBs to replace the square convolutional kernel in the UNet. For a square convolution kernel of size $3 \times 3$, we replace it with 3 convolution blocks of size $3 \times 3$, $3 \times 1$, and $1 \times 3$. ACBs add horizontal and vertical asymmetric convolution kernels based on a square convolution kernel, which increase the receptive field and improves the network performance.

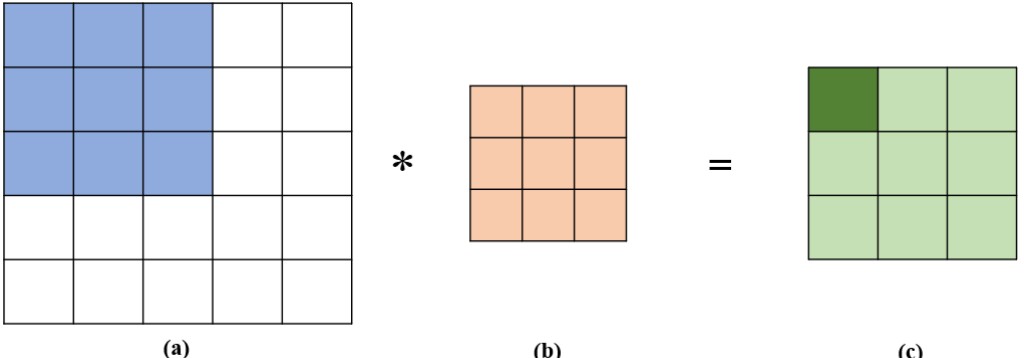

**Figure 6.** Schematic diagram of convolution operation: (**a**) the input matrix, which is acquired data matrix in this paper; (**b**) convolution kernel matrix; (**c**) the convolutional result matrix. The $*$ represents convolution operation.

As shown in Figure 7, the ACBs comprise three parallel layers, and the outputs of three parallel layers are summed up as the output of the ACBs. Due to the addition of convolutions with compatible kernel sizes, the ACBs' asymmetric convolution kernels can be equivalently fused into a standard square kernel [38,39]. The processing workflow can be expressed as follows (5):

$$I * K = I * K^{(1)} + I * K^{(2)} + I * K^{(3)} = I * (K^{(1)} + K^{(2)} + K^{(3)}) \tag{5}$$

where $I$ denotes the input matrix; $K^{(1)}$, $K^{(2)}$, and $K^{(3)}$ are 2D kernels with compatible sizes, denoting $3 \times 3$ square convolution kernel, $1 \times 3$ asymmetric convolution kernel, and $3 \times 1$ asymmetric convolution kernel, respectively; $\oplus$ is the element-wise addition of the kernel parameters on the corresponding positions; and $K$ is the equivalent convolution kernel.

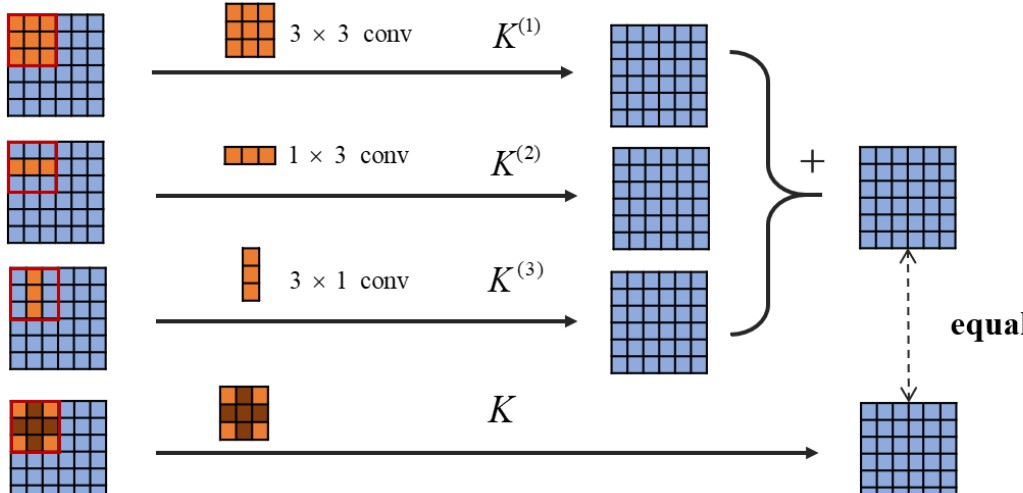

**Figure 7.** Schematic illustration of the ACB.

## 4. Numerical Results

### 4.1. Synthetic Data

To test the reliability of the trained ACB-UNet, we use it to attenuate the traffic noise in the synthetic data. Figure 8a shows synthetic data without noise, the number of data traces was 360, and the record length was 7 s. Figure 8b shows the noise we extracted from the real data, which mainly includes two types of typical traffic noise: noise distributed in the fan-shaped and thin strip. The synthetic data containing noise is shown in Figure 8c, from which we can see that the road noise covers the reflection and affects the analysis and utilization of the effective signal. We further analyze the characteristics in f-k spectra, as shown in Figure 8d. The reflection is mainly distributed in the range of 0–20 Hz. Noise is primarily distributed in the range of 3–60 Hz, coupled with reflection in the 3–20 Hz range.

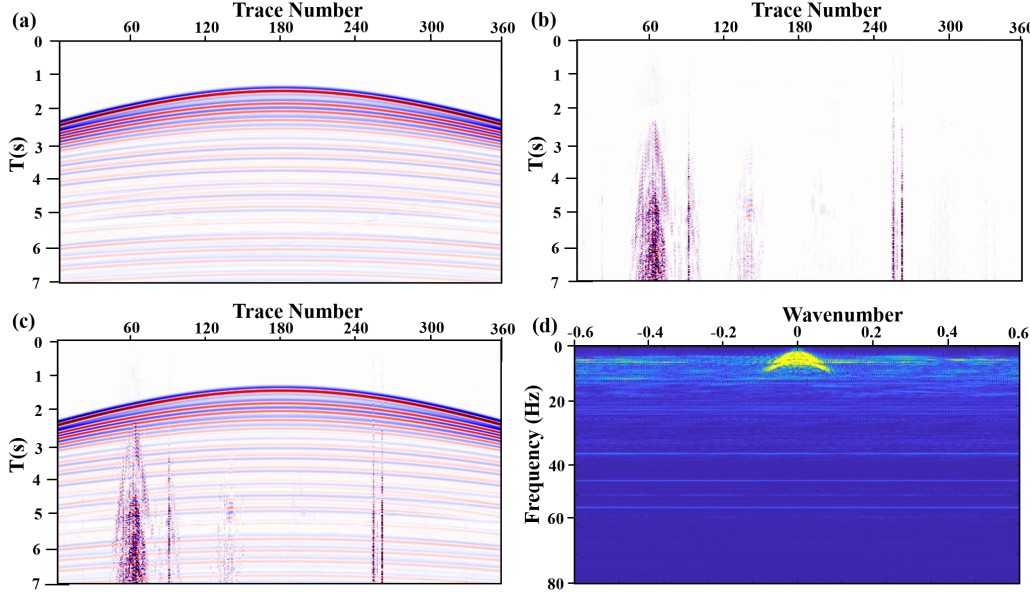

**Figure 8.** (**a**) Synthetic data without noise; (**b**) traffic noise; (**c**) synthetic data after adding noise to (**a**); (**d**) the f-k spectra of (**c**).

We use the proposed ACB-UNet method to attenuate traffic noise in synthetic seismograms and compare it with other methods. Figure 9a,b show the denoising results and noise section obtained by f-k filtering. It can be seen from Figure 9a that there are still noise residues in the denoised records, and the reflection at the noise location has been distorted. From Figure 9b, we can see serious artifacts in the noise section caused by the F-K algorithm mechanism. Figure 9c shows the denoised result obtained by the UNet. The thin strip noise has been well removed, but the fan-shaped noise (near the 60th trace) removal is not ideal, and there is still noise residue. The result after the ACB-UNet attenuates the traffic noise is shown in Figure 9e, and there is no longer visible noise residue in the denoised result.

Figure 10 shows the f-k spectra of Figures 8a and 9e. Comparing Figures 8d and 10a, it can be found that there is no aliasing in the f-k spectra of clean records. As can be seen in Figure 10a,b, there is no significant difference between the f-k spectra of the two, and the noise is effectively attenuated. The trained ACB-UNet method can recover the f-k spectra. To quantitatively evaluate the performance of the ACB_UNet method in synthetic data, we plotted the amplitude spectra, as shown in Figure 11. The blue line corresponds to the synthetic data with traffic noise, the black line represents the synthetic data, and the red line denotes the results obtained by the ACB_UNet. Compared to the lines, the energy of traffic noise is successfully attenuated by the ACB_UNet. Figure 11b shows the enlarged rectangular area. The result obtained by the ACB_UNet is very close to the synthetic data, with no significant loss of effective signal.

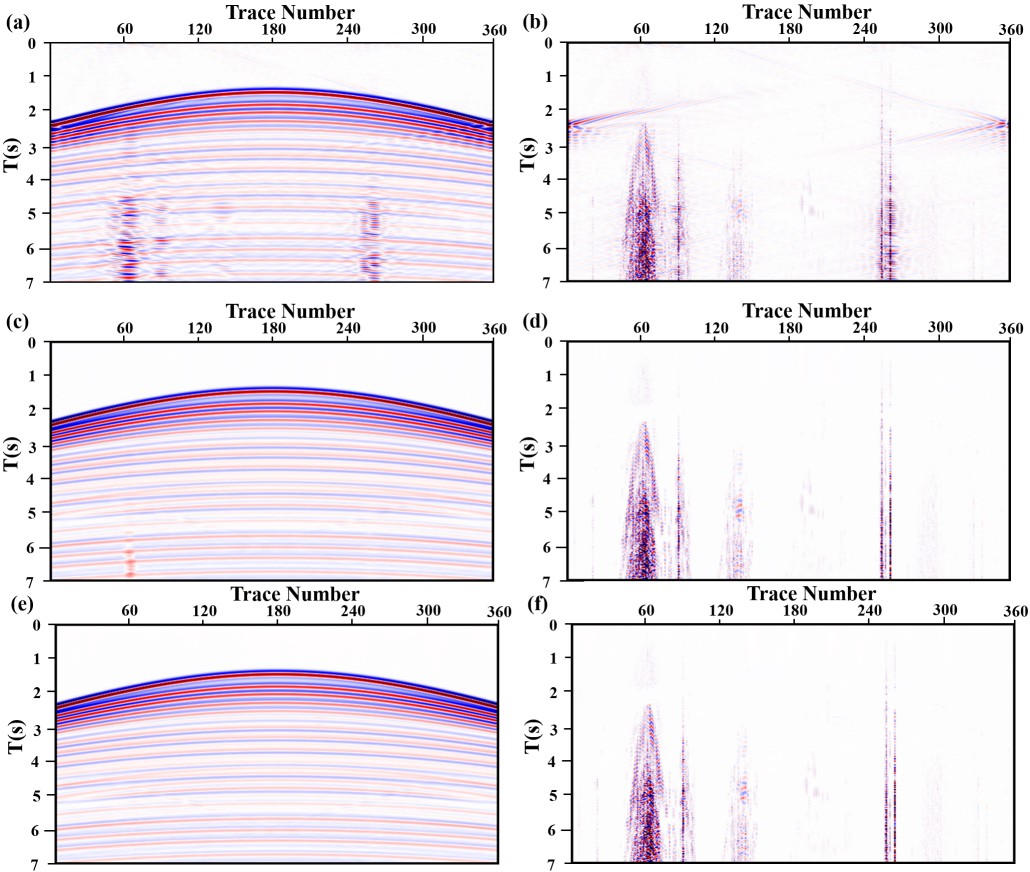

**Figure 9.** The denoising results and noise sections of different methods for synthetic data: (**a**) the denoising result and (**b**) noise section using f-k filtering; (**c**) the denoising result and (**d**) noise section using the UNet; (**e**) the denoising result and (**f**) noise section using the ACB-UNet.

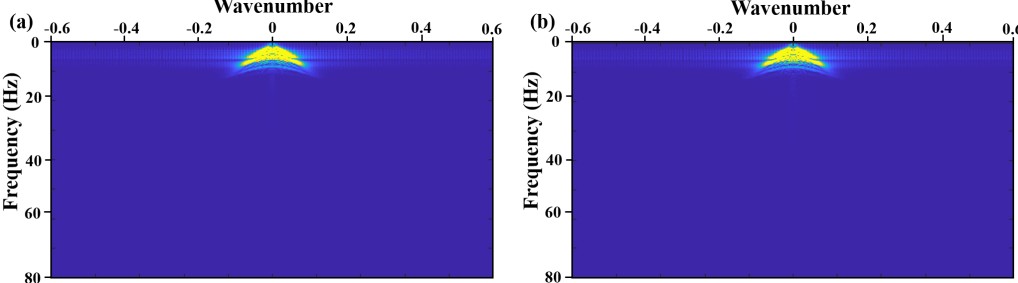

**Figure 10.** The (**a**) f-k spectra of the data in Figure 8a; (**b**) f-k spectra of the data in Figure 9e.

### 4.2. Real Data

In this section, we verify the validity of the ACB-UNet on real data. As the geophones are deployed near the highway, the real data are heavily contaminated by traffic noise, and the noise has a variety of forms. We further prove the effectiveness of the ACB-UNet method by attenuating the noise in real data.

Figure 12 shows a 3D pre-stack seismic record, which is not fed to the network for training. The record has 360 traces, and the trace interval is 25 m. Each trace has 3501 samples with a time interval of 2 ms. As seen from the figures, the traffic noise is most serious near traces No. 60 and No. 120, which are distributed in a fan shape and cover the reflections of the data.

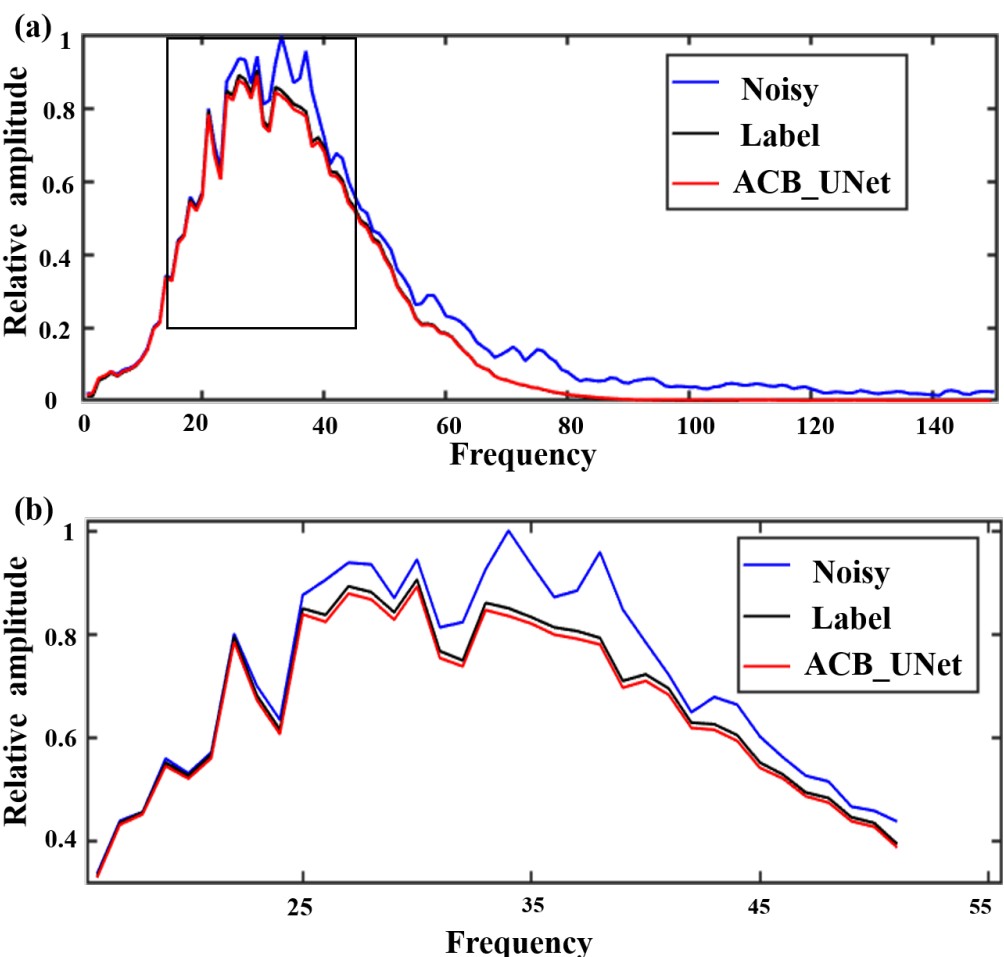

**Figure 11.** (**a**) Amplitude spectra of Figures 8a,c and 9e; (**b**) enlarged rectangular area in (**a**).

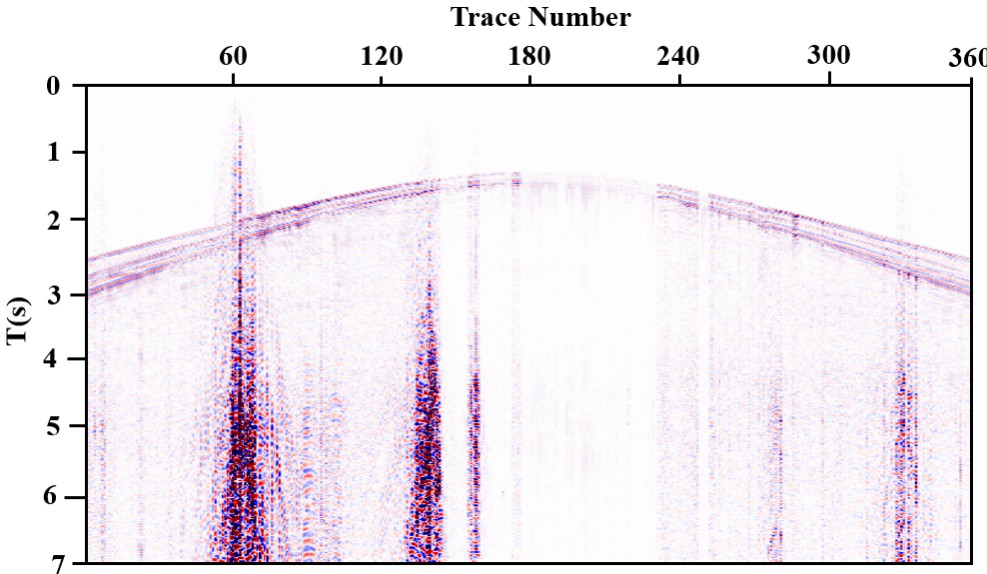

**Figure 12.** The original data with traffic noise.

To attenuate the traffic noise, we apply f-k filtering, the UNet, and the ACB-UNet methods to the data. Figure 13a shows the data obtained using f-k filtering to the original data. We can see that the denoising result of f-k filtering is not satisfactory, and abundant

residual noise still exists in the record. Figure 13b shows the noise section processed with f-k filtering. There is some signal leakage to the noise section, especially since the effective signal loss near the first break is serious. The denoising results of the UNet are shown in Figure 13c, and Figure 13d is the noise section. The method effectively eliminates traffic noise, and the covered reflections are restored. However, there are some artifacts in the denoising results of the UNet (as shown by the red arrow in Figure 13c), and its accuracy is preferable to further improvement. Figure 8e shows the results processed by the ACB-UNet. Compared with other methods, the denoising result of the ACB-UNet has almost invisible traffic noise and no errors such as artifacts. For further analysis and comparison, we enlarge the rectangular area in Figure 13c,e to obtain Figure 14a,b. As shown in Figure 14a, the denoising results of the UNet still have relatively serious artifact problems (as shown by the black arrows). Comparing the area marked by the blue arrows, we can see that the ACBs remove noise while effectively retaining the signal. It has better detailing capabilities than the UNet, demonstrating the advantage of an asymmetric convolution kernel.

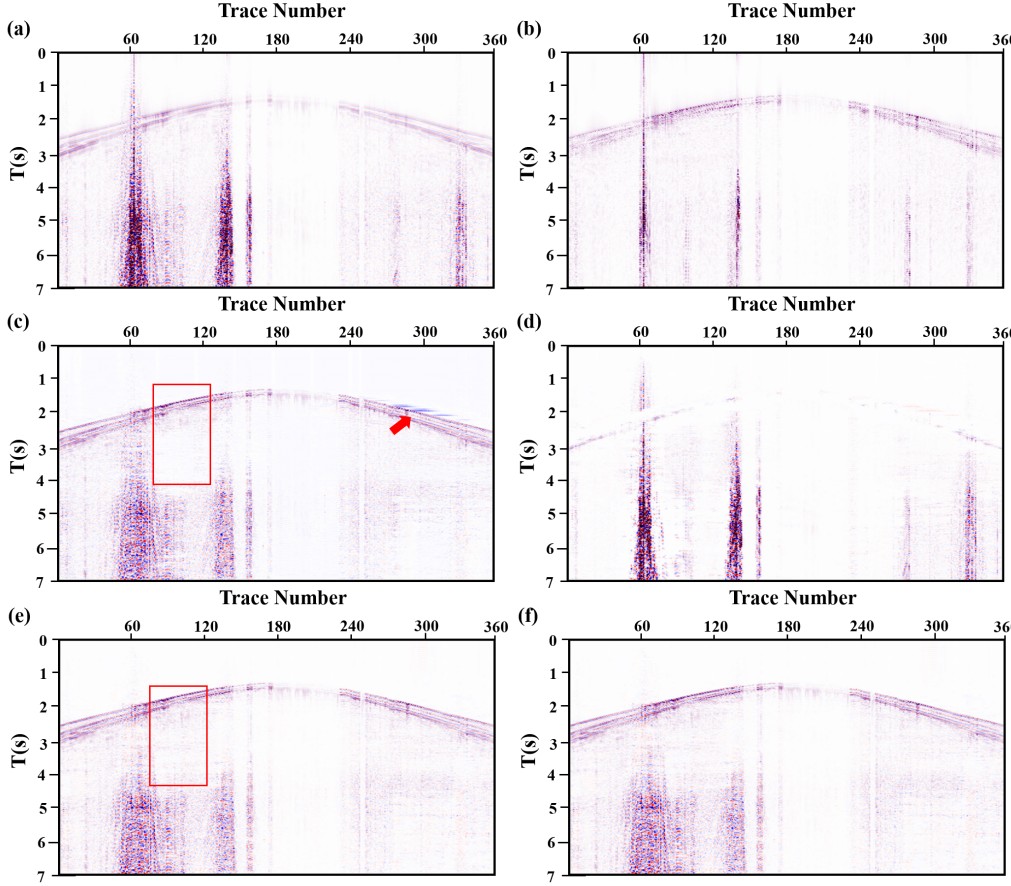

**Figure 13.** The denoising results and noise sections of different methods for real data: (**a**) The denoising result and (**b**) noise section using f-k filtering; (**c**) the denoising result and (**d**) noise section using the UNet; (**e**) the denoising result and (**f**) noise section using the ACB-UNet.

We analyze the data before and after denoising on the f-k spectra. Figure 15a,b show the f-k spectra of the original data and the ACB-UNet denoising result. Figure 15 shows that the traffic noise overlaps with reflections in 0–30 Hz, and the traffic noise is strong in energy, covering almost all of the effective signals. It can also be inferred from this figure that the noise overlapped with the signal, leading to the unsatisfactory denoising effect of the f-k filter. Figure 15b shows the f-k spectrum of the data in Figure 13e. The traffic noise is invisible in the f-k spectra of the data processed by the ACB-UNet. It is evident that the ACB-Net denoising performance is better than other methods.

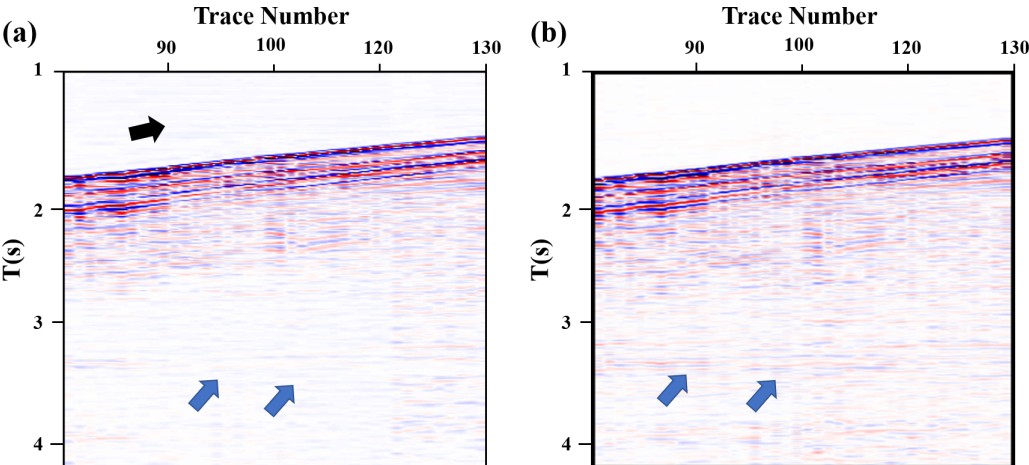

**Figure 14.** The enlarged rectangular area of (**a**) the UNet and (**b**) the ACB-UNet denoising results.

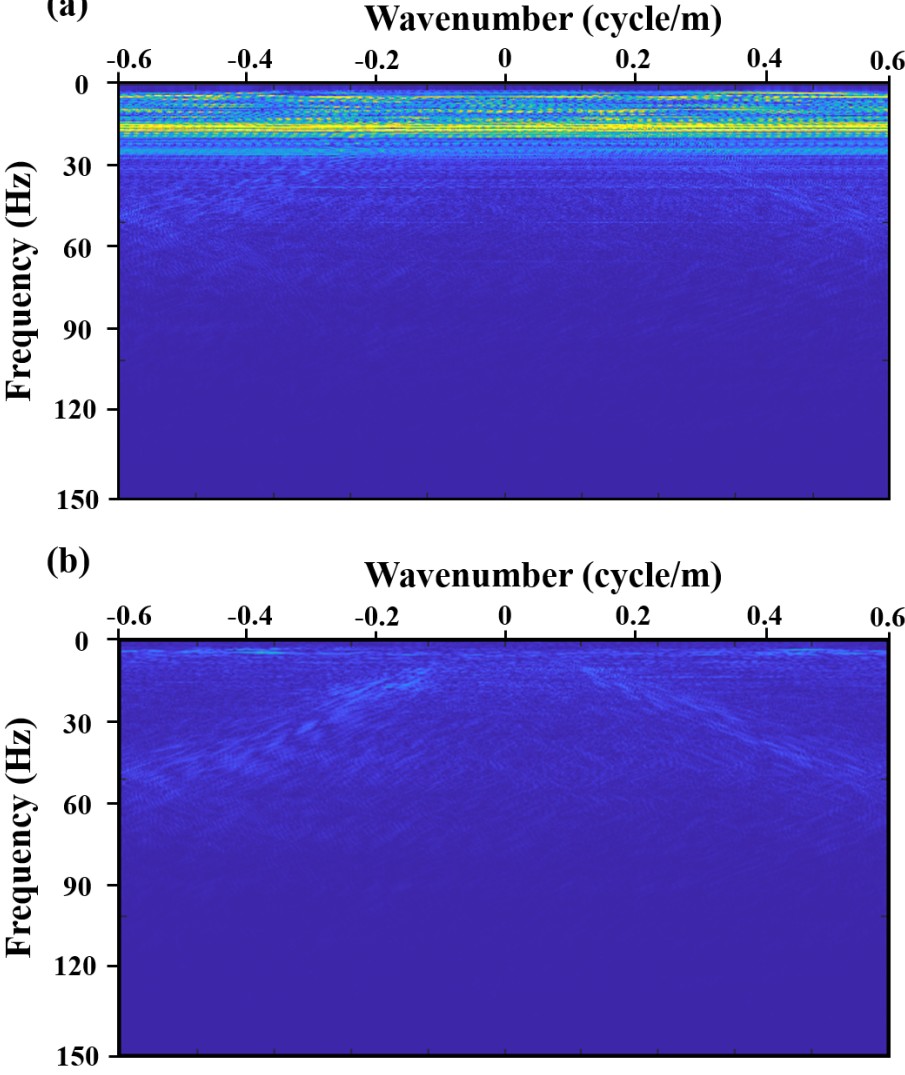

**Figure 15.** (**a**) The f-k sepectrum of original data; (**b**) the f-k sepectrum of the ACB-UNet denoised data.

## 5. Conclusions

In this paper, we analyzed the characteristics of traffic noise in seismic records and proposed a combination of the UNet and ACBs to achieve attenuation of traffic noise in seismic or acoustic records. In the real data, high amplitude traffic noise covers the signal and affects the data processing and interpretation. Therefore, traffic noise attenuation is an essential step. However, the traffic noise and the signal significantly challenge noise attenuation work. Unlike traditional denoising methods, the UNet uses training data to extract potential features between data and can realize intelligent denoising without any assumptions and domain transformation. The ACBs use $3 \times 1$, $1 \times 3$, and $3 \times 3$ convolution kernels instead of a $3 \times 3$ square convolution kernel in the UNet, which can be easily integrated into the the UNet framework to improve network feature extraction and detail processing capabilities. Another significant advantage of the method is that the cost lies mainly in network training. The prediction after successful network training is very convenient and does not require any parameter adjustment. The trained network can learn the potential features between the data and identify and attenuate the traffic noise in the data. This method has the potential to identify and separate different data as long as we build rich training samples for the data. Synthetic and real data tests show that the method can effectively attenuate high amplitude traffic noise in seismic records, improve the signal-to-noise ratio of seismic records, and achieve effective utilization of seismic data. At the same time, the method preserves effective signals, especially those with weak energy, while attenuating road traffic noise. Moreover, the method has favorable application foreground.

**Author Contributions:** Conceptualization, Z.Z. and D.C.; methodology, Z.Z., X.C. and D.C.; validation, X.C., M.C. and S.D.; writing—original draft preparation, Z.Z. and X.C.; writing—review and editing, Z.Z. and X.C.; funding acquisition, Z.Z. All authors have read and agreed to the published version of the manuscript.

**Funding:** The research was funded by Hainan Provincial Natural Science Foundation of China, Grand No.: 422MS091, and funded by Hainan Provincial Joint Project of Sanya Yazhou Bay Science and Technology City, Grand No.: 2021CXLH0018, and funded by Hainan institute Foundation of Zhejiang University.

**Institutional Review Board Statement:** Not applicable.

**Informed Consent Statement:** Not applicable.

**Data Availability Statement:** The data are available from the corresponding author upon request.

**Conflicts of Interest:** The authors declare no conflict of interest.

## Abbreviations

The following abbreviations are used in this manuscript:

| | |
|---|---|
| CNN | convolution neural networks |
| ACB | asymmetric convolution blocks |
| ACB-UNet | U-Net with asymmetric convolution blocks |

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
