# Peer review of "U-Net with Asymmetric Convolution Blocks for Road Traffic Noise Attenuation in Seismic Data"

_applsci, doi:10.3390/app13084751_

Round 1

Reviewer 1 Report

The originality of the manuscript is essentially in the application; the technical is otherwise not new, as it uses known algorithms and follows standard methodology. There are so far two main criticisms:

1) The problem of denoising data has been studied for long in signal and image processing. Many efficient techniques have been proposed. They are not all reviewed in the paper. One may wonder why another method is then needed and, most importantly, if it can do better than the state of the art.

2) Since the denoising algorithm is applied on synthetic data, it remains a "simple" school exercise. It seems necessary to validate it on real data, in particular in the presence of traffic noise that has not been used in the training stage.

Author Response

We thank the reviewers for taking the time to review our manuscript and give their constructive comments and helpful suggestions. These professional comments/suggestions are of great help to the improvement of the quality of this manuscript. The manuscript has been revised carefully and thoroughly to address the issues raised in the reviews.

1) The problem of denoising data has been studied for long in signal and image processing. Many efficient techniques have been proposed. They are not all reviewed in the paper. One may wonder why another method is then needed and, most importantly, if it can do better than the state of the art.

Response 1:  Accepted and done. We have added a description of traditional denoising methods to the manuscript. At the same time, in the test of synthetic data and actual data, it can be found that the proposed method does not require manual operation, can effectively achieve noise attenuation, and can reduce the loss of effective signals.

2) Since the denoising algorithm is applied on synthetic data, it remains a "simple" school exercise. It seems necessary to validate it on real data, in particular in the presence of traffic noise that has not been used in the training stage.

Response 2:   In order to prove the effectiveness of the method proposed in this article, in Section 4.2, we conducted a traffic noise attenuation test on actual data, and obtained ideal results.

Reviewer 2 Report

1) In my opinion the title of the article needs improvement first of all because 'Road Traffic Noise Attenuation' usually is understood as a task for one type of environmental noise which is radiated and propagates in the atmosphere. Even the abstract does not reveal the main essence of the article - what is meant by road noise, which spreads in the form of surface waves and can be masking for the study of seismic (acoustic) phenomena.

2) In the same context it is difficult to understand the introduction: 'Ground roll, Traffic noise in land survey and swell noise in marine acquisition are the very important ambient noises which have the high amplitude and mixed models. Removing one of these noises  plays a key role for the quality of final results.' - Land survey for what purposes? The quality of final results for what purposes?

3) Next sentence in Introduction - 'The road noise are used to demonstrate our methodology in this paper' - leaves no doubt that it is not about road noise фttenuation, but road noise is chosen as an example of the implementation of the developed method presented in the article.

4) In Conclusions the first sentence 'This paper proposes a combination of UNet and ACB for seismic or acoustic data traffic noise attenuation' fully confirms that traffic noise in the article is a masking noise for the signal which important to investigate.

5) Final conclusion once again is talking about 'handling details much better' as the main result, but once again for what?: This method has the potential to identify and separate different data as long as we build rich training samples for the data. Synthetic and real data tests show that the method can effectively attenuate high-amplitude traffic noise and handle details much better, which is promising.

6) In my suggestion the article in current conditions missed something important - it does not talk about the main purpose - for what the method is used?

Author Response

We thank the reviewers for taking the time to review our manuscript and give their constructive comments and helpful suggestions. These professional comments/suggestions are of great help to the improvement of the quality of this manuscript. The manuscript has been revised carefully and thoroughly to address the issues raised in the reviews.

1) In my opinion the title of the article needs improvement first of all because 'Road Traffic Noise Attenuation' usually is understood as a task for one type of environmental noise which is radiated and propagates in the atmosphere. Even the abstract does not reveal the main essence of the article - what is meant by road noise, which spreads in the form of surface waves and can be masking for the study of seismic (acoustic) phenomena.

Response 1: Accepted and done. We have modified the title. And we have provided a more detailed description of the generation of traffic noise in the abstract, so that readers can more clearly understand the causes of traffic noise.

2) In the same context it is difficult to understand the introduction: 'Ground roll, Traffic noise in land survey and swell noise in marine acquisition are the very important ambient noises which have the high amplitude and mixed models. Removing one of these noises plays a key role for the quality of final results.' - Land survey for what purposes? The quality of final results for what purposes?

Response 2:  Accepted and done. We have corrected the relevant description in the manuscript.  Land survey mainly involves obtaining seismic records of a certain area, performing inversion and imaging operations on the seismic records to obtain the structure of the underground medium in the area, and using it for subsequent oil and gas exploration. Noise in seismic records can reduce the quality of seismic data and make full use of effective signals. The denoising of seismic data is conducive to improving the signal to noise ratio of seismic records and facilitating subsequent inversion, imaging, and other operations.

3) Next sentence in Introduction - 'The road noise are used to demonstrate our methodology in this paper' - leaves no doubt that it is not about road noise фttenuation, but road noise is chosen as an example of the implementation of the developed method presented in the article.

Response 3: Accepted and done. The expression here may not be clear, and we have made corresponding changes in the manuscript.  

4) In Conclusions the first sentence 'This paper proposes a combination of UNet and ACB for seismic or acoustic data traffic noise attenuation' fully confirms that traffic noise in the article is a masking noise for the signal which important to investigate.

Response 4:  Traffic noise is the mixture of harmonic noise and ground roll from passing vehicles, which are mixed with valid signals in seismic records, making seismic data more complex. Traditional methods are difficult to achieve a balance between denoising and preserving valid signals. At the same time, its energy is much greater than the signal, which can mask some effective signals and affect the effective utilization of the signal. To this end, we studied the characteristics and distribution of traffic noise, using the deep learning methods to attenuate traffic noise in seismic data.

5) Final conclusion once again is talking about 'handling details much better' as the main result, but once again for what?: This method has the potential to identify and separate different data as long as we build rich training samples for the data. Synthetic and real data tests show that the method can effectively attenuate high-amplitude traffic noise and handle details much better, which is promising.

Response 5: Accepted and done. The energy of seismic traffic noise is greater than the effective signal, which can mask the effective signal and affect the use of seismic data. When denoising, we should not only focus on the removal of noise, but also on the quality of the data after denoising. Compared with other methods, the method proposed in this paper can avoid the loss of effective signals while removing noise, preserve the details of effective signals, and facilitate subsequent use.

6) In my suggestion the article in current conditions missed something important - it does not talk about the main purpose - for what the method is used?

Response 6: Accepted and done. We add a corresponding description in the conclusion, which shows the main purpose of the method.